# Niche-Driven Socio-Environmental Linkages and Regional Sustainable Development

**Dandan Liu [1], Anmin Huang [1,*], Dewei Yang [2], Jianyi Lin [3] and Jiahui Liu [3]**

1    College of Tourism, Huaqiao University, Quanzhou 362021, China; 00_liu@sina.cn
2    School of Geographical Sciences, Southwest University, Chongqing 400715, China; younglansing@gmail.com
3    Key Lab of Urban Environment and Health, Institute of Urban Environment, Chinese Academy of Sciences, Xiamen 361021, China; jylin@iue.ac.cn (J.L.); jhliu@iue.ac.cn (J.L.)
*    Correspondence: amhuang@hqu.edu.cn

**Abstract:** The changes in niche roles and functions caused by competition for survival resources have implications in various domains, with natural science and social science standing out. Currently, expanding the ecological niche concept and its practical interpretation in the fields of social ecology, geography and sustainable science is becoming a crucial challenge. This paper is based on niche theory to observe niche evolution and resulting socio-ecological effects of 1186 towns in 19 prefecture cities in Yangtze River delta. The results indicate that: Towns around the Taihu Lake displayed obvious spatial agglomeration, which was leading the development of the entire region. The town niche shows obvious characteristics of north-south differences and hierarchy distribution. The niche coordination degree of Jiangsu Province was higher than that of Zhejiang Province. The higher the subsystem coordination degree, the better the town development. Towns with poor ecological conditions are often subject to competition, while towns with better ecological conditions often benefit from cooperative development. The niche separation and collaboration could enhance niche competition of towns and cities in the region. The proposed framework can facilitate interdisciplinary exchanges among geography, sociology, landscape ecology and regional planning and provide insights for understanding regional co-opetition relationship and regional sustainable development.

**Keywords:** niche; spatial pattern; co-opetition relationship; regional regulation; towns

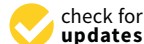



## 1. Introduction

A city is a coupled system that is composed of natural, social and economic components [1,2]. The city brings us convenience in life while a certain number of unexpected so-called "city diseases", e.g., excessive resource consumption, ecosystem damage, and environmental pollution. These urban health problems, caused by rapid urbanization and intensive industrialization, not only exert pressure on natural system sustainability, but also result in socio-economic system risks, e.g., regional disparity, social inequity, non-nutritive culture, and institutional constraints [3,4]. These increasing challenges have been set priority of research issues on regional competition and differentiation.

During recent years, it has been noticed that rapid urbanization is accompanied by complicated challenges in terms of materials, energy and information issues. These challenges need to be comprehensively investigated by observing multiple factors [5–7]. The early research methods of socioecological system mainly include evaluation methods such as analytic hierarchy process, ecological footprint analysis and grey model [8–10]. In the later stage, scholars focused on the application of models and theories, such as DPSIR model [11], Bayesian network [12], full permutation polygon synthetic indicator [13], ArcGIS participation mapping [14], niche and so on. Rooted in the fields of ecology and biogeography, ecological niche refers to the spaces and roles of species in biological community [15] which indicates the interactive relationships between a certain species with the outside environment [7,16–18]. Similarly, multifactor approaches in urban studies have



also benefited from ecological niche theory [6,19,20]. Urban niche refers to the position, function and role of a city in a region and the developing resources obtained by a city that cover natural, economic and social aspects within and beyond the city boundary [13,21]. Urban niche as an emerging, and interdisciplinary perspective allows us to understand socio-ecological processes in human-dominated systems and help societies reach sustainable development goals [22,23]. Towns, as the base and nodes of the urban system pyramid, play a unique role in the rise of China's cities. Thousands of Chinese towns foster gradient cities by supplies of materials, energy and information flows. The introduction of ecological niche theory into the study of towns help understand the development potentials of a city on a more micro scale and find a competitive niche.

Ecological niche has been an important concept in ecology [24]. Ecological niche can be dated back to the application of Johnson. Grinnell is credited with its formal development and his original formulation of niche as a spatial unit [25–27]. Elton conceived of niche as a functional unit [28,29]. Gause discovered the competitive relationship between species [30]. Hutchinson inspired from both Grinnell's and Elton's niche definitions, redefined niche as an "*n*-dimensional hypervolume" [18,31]. Odum emphasized the adaptability of species under the influence of internal and external environmental factors [32]. Most current works in niche theory are associated with Hutchinson's multidimensional niche [26]. In modern ecology, niche theory plays an increasingly important role in the study of interspecific relationship, community structure, species diversity, and species evolution [16,33,34].

In recent years, the concept of niche in ecology has been borrowed to the research of social sciences, such as economics, human ecology, sociology, management science, geography and sustainability science [34,35], and is being widely used. Similar to the natural world, the niche of human society reflects the roles and functions of social units. Broussard and Young considered with the addition of culture, human niches are even more multidimensional than those of other organisms [26]. Beyond that, occupational niche, cultural niche and organizational niche are also attempting to combine human niche with natural niche theory [36]. Therefore, the complex connections between human activities and natural systems make them constitute a social-ecological feedback system used ecological niche theory to consider the effects of improvements in health, agriculture, or efficiency on the abundances of human resources applied an eco-cultural niche model to illuminate the past human-environment interaction [37,38].

Looking back in history, we can find that the human society development is actually a process of continuous social differentiation, continuous exploitation of resources, and continuous expansion of living space [39]. Therefore, the priority fields of human niche research should be the space and resources which human survival depends on, especially in urban niche [1,40,41]. Many scholars have made important contributions to the formation and development of urban niche theory from a diversity of perspectives. Wang and Han considered that city occupies a niche in a natural-economic-social complex ecosystem [13]. Jiang pointed out that an ecological niche involves many elements, such as resources, environment, economy, and society [21]. Salvati proposed a composite index of urban complexity with the aim to outline the inherent spatial gradient associated with settlement morphology, social diversification, local development, and economic performance [6]. Most of these studies evaluate the urban niche from the perspective of the comprehensive eco-economic system, of which the 'niche' concept has been applied to the urban fields, e.g., post-industrial city development [20], economic structure, globalization and information technology, and ecological sustainability [42].

From the perspective of research methods, the concept of niche is abstract and fuzzy. What we can learn from it is some quantitative indicators. such as niche width, niche overlap, niche volume and niche dimension. Niche breadth and niche overlap are the most important quantitative indicators. Levins was the first to put forward the measurement formula of niche width, and also have Schoener, Hurlbert, Smith, etc [43]. The formulas of niche overlap are curve average method, symmetry $\alpha$ method [44] and so on. At present, the theory of niche is still in continuous development, and various measurement formulas

are constantly improving, especially in the quantitative calculation of niche width. Each formula has its advantages and disadvantages. It can be predicted that, with the continuous improvement of niche theory, it will inject more fresh blood into the theory and method of urban research.

Overall, research gaps remain. First, the urban niche theory allows us to observe the functions and roles of cities and towns in their "ecosystem", and uncover the city-related influencing factors for niche competition. Currently research mainly focused on the macroscale region or city, lacking the detailed study of microscale towns. Therefore, a new perspective is proposed to study the functional linkages and system roles of towns using the niche theory. Second, it is urgent to establish the ecological niche evaluation index system for investigating town niche within a connected regional system. There is still a lack of a complete indicator system, employing big data on the national, provincial and municipal scale, to overall study the towns as the basic cell of social and economic development. Moreover, a lot of room remains for discussion on the standards of indicators. Third, the current niche methods are mainly based on niche breadth and overlap. More diversified research methods are needed to enrich niche theory for uncovering meaningful results. Therefore, the spatial analysis of ecological niche combining a toolbox of ArcGIS, SPSS and indicator system attempts to discuss the spatial coordination development and regional resource flows.

Inspired by the niche theory, a modified niche-based comprehensive evaluation framework of ecological niche will be established to quantitatively analyze the development status, spatial differences, and future development strategies at the town scale. This provides new insights for the analysis of spatial patterns, regional transformation and sustainable development from the niche perspective at town scale. This paper attempts to: (1) Establish a modified niche-based comprehensive evaluation framework of ecological niche for better understanding the town ecological niche in a regional ecosystem. This framework consists of three-aspect index system, niche-related methods, and spatial analysis methods, which depicts the elements dynamics and spatial pattern of resources, environment, economy, and society; (2) observe the ecological niche characteristics of 1186 towns in terms of spatial patterns and interactive relationships in Zhejiang and Jiangsu provinces in 2017. Understanding the co-benefits and negative effects during the competitive and cooperative process could help identifying developing potentials of the towns; (3) provide insights for functional plans and complementary development measures from the microscale towns to the macroscale cross-boundary provinces. These suggestions could be extracted from factorial analysis and niche changes in each town, e.g., pollution index and fitness index [45].

## 2. Materials and Methods

### 2.1. Study Area

The Yangtze River Delta (YRD) region, located in the eastern coastline of China, is experiencing rapid economic development and land urbanization in recent decades [46]. Jiangsu Province and Zhejiang Province in the YRD region, which ranked second and fourth in China in terms of regional gross domestic product (GDP), in 2017, are selected to the evolution of regional niche. Looking back into the rise of the study area, towns relying on private industries in two provinces play a prominent role in both environmental and economic fields. It is also a region with deep concern of regional integration and co-opetition relationships because of the historically rooted north-south divide [47]. The evolution of town niche could provide a reference for sustainable towns management. The paper selected 1186 towns in Jiangsu Province and Zhejiang Province, without consideration of sub-district office due to urbanization area. Thus, the non-study area is not the research object in this paper. The location and basic information of case study are shown in Table 1 and Figure 1.

**Table 1.** Basic information of case study.

| Province | City | Population (Million) | Area (km²) | GDP (Billion Yuan) | Number of Towns |
|---|---|---|---|---|---|
| Jiansu | Nanjing(J1) | 8.33 | 6587.02 | 1171.51 | 13 |
| | Wuxi(J2) | 6.55 | 4627.47 | 1051.18 | 27 |
| | Suzhou(J3) | 10.68 | 8657.32 | 1731.95 | 52 |
| | Nantong(J4) | 7.31 | 8544.00 | 773.46 | 60 |
| | Yangzhou(J5) | 4.51 | 6591.21 | 506.49 | 64 |
| | Zhenjiang(J6) | 3.19 | 3847.00 | 410.54 | 31 |
| | Taiizhou(J7) | 4.65 | 5787.26 | 474.45 | 78 |
| | Changzhou(J8) | 4.72 | 11,765.00 | 662.22 | 36 |
| Zhejiang | Hangzhou(Z1) | 9.47 | 16,853.57 | 1255.62 | 90 |
| | Ningbo(Z2) | 8.01 | 9816.00 | 984.69 | 84 |
| | Wenzhou(Z3) | 9.22 | 11,612.94 | 545.32 | 63 |
| | Huzhou(Z4) | 2.99 | 5820.13 | 247.61 | 39 |
| | Shaoxing(Z5) | 5.01 | 8274.79 | 510.80 | 88 |
| | Jiaxing(Z6) | 4.66 | 4275.05 | 435.52 | 40 |
| | Jinhua(Z7) | 5.56 | 10,941.42 | 387.02 | 106 |
| | Quzhou(Z8) | 2.18 | 8844.79 | 138.00 | 76 |
| | Zhoushan(Z9) | 1.17 | 22,200.00 | 121.90 | 22 |
| | Lishui(Z10) | 2.19 | 17,275.00 | 129.82 | 135 |
| | Taizhou(Z11) | 6.12 | 10,050.43 | 438.82 | 82 |

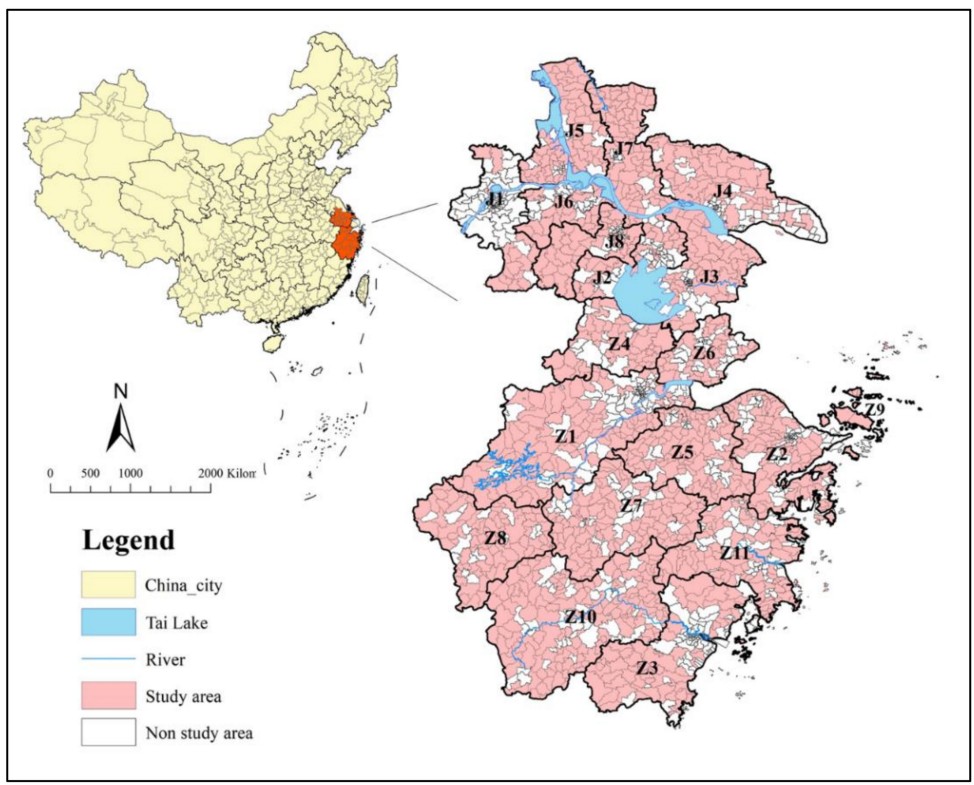

**Figure 1.** Location of study area.

*2.2. Methods*

In this study, a niche-based comprehensive evaluation framework consisting of a three-layer niche-based evaluation index system, three niche-related methods and three spatial analysis methods was established to observe the dynamics and spatial differences of town niche. First, a three-layer niche-based evaluation index system was constituted, including target layer, sub-target layer (environment, economic, and societal niche), and indicator Layer (14 indicators). This index system can help identify the environment, economic, and

societal niche by categorizing developing indicators. Second, the niche-related methods, i.e., the degree of niche breadth, niche overlap, and coordination, were used to analyze the developing potentials and co-opetition relationships of town niche. Third, three spatial analysis methods, i.e., spatial autocorrelation analysis, Pearson correlation coefficient, and gravity model, were employed to evaluate spatial interaction and differences of town niche. Furthermore, understanding the co-benefits and negative effects during the competition and cooperation on niches could help identifying developing potentials of the towns (Figure 2).

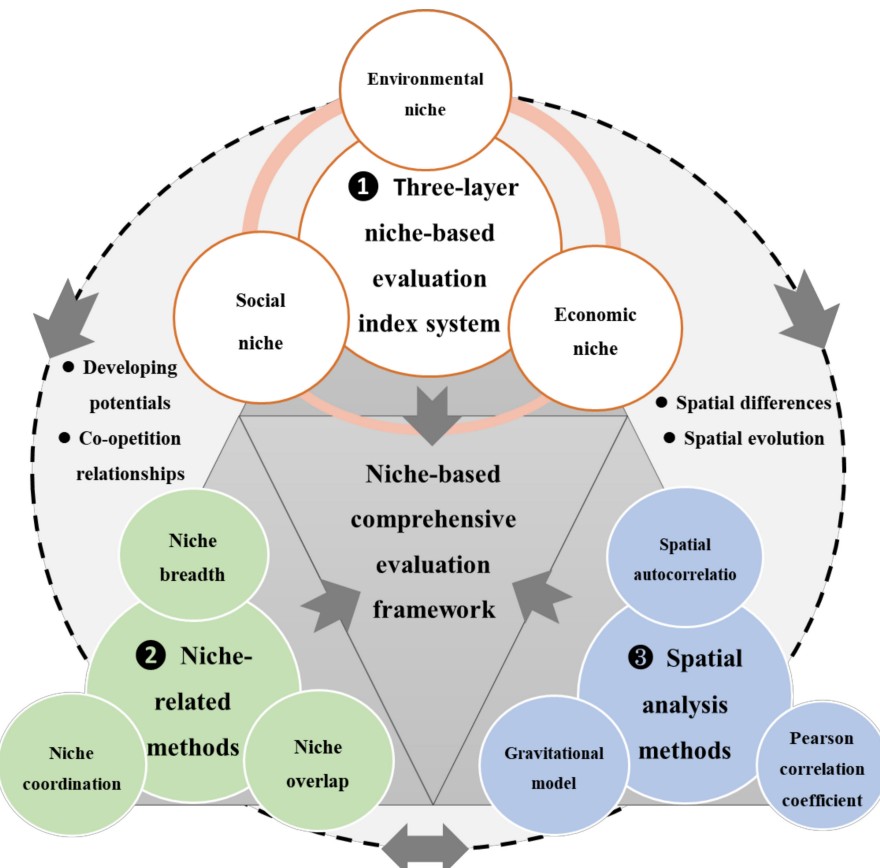

**Figure 2.** Research flowchart in this study.

### 2.2.1. Niche-Based Evaluation Index System

A town is essentially a complex ecosystem, the environment, resource and social system as supporting factor. Therefore, analysis of niche involves many elements such as environment, economy and society [2,13]. Towns are very small in China's administrative divisions, resulting in incomplete data. The index system of this study is based on the availability of data, combined with the foreign and domestic niche and sustainable development index system. According to different town functional modules and incorporating the system of suitable town indices, the target layer is the comprehensive ecosystem, the sub-target layer includes the niches of the environment, economy, and society, and the indicator layer comprises the individual indicators for the corresponding niches. The environmental niche reflects the advantages and disadvantages of the region in resource endowments in and natural ecosystem protection and restoration, includes water area, wetland area, normalized difference vegetation index, and Particulate Matter 2.5 (PM2.5). The economic niche indicates the advantages and disadvantages of the region in terms of economic developments, includes four indicators that reflect the economic scale and economic structure of the region. The social niche reflects the advantages and disadvantages of public facilities and welfare in a region, includes six indicators that reflect the

advantages and disadvantages of a region regarding education, medical service, health, social security, quality of life, and rural living facilities. The PM2.5 is a negative indicator, the smaller, the better (Table 2).

**Table 2.** Social-economic-environment niche evaluation index system.

| Target Layer | Sub-Target Layer | Indicator Layer | Index Weight |
|---|---|---|---|
| Comprehensive ecological niche of towns | Environmental niche | Water area ($m^2$) | 0.02 |
| | | Wetland area ($m^2$) | 0.05 |
| | | Normalized difference vegetation index ($m^2$) | 0.01 |
| | | Particulate Matter 2.5 ($\mu g/m^3$) | 0.02 |
| | Economic niche | Total industrial output per capita (10,000 yuan/person) | 0.06 |
| | | Per capita financial income (10,000 yuan/person) | 0.13 |
| | | Percentage of industrial enterprises in the number of enterprises (%) | 0.05 |
| | | Catering, accommodation, shopping and entertainment facilities (number) | 0.06 |
| | Social niche | Transportation facilities (number) | 0.13 |
| | | Stadiums (number) | 0.19 |
| | | Parks and leisure plazas (number) | 0.07 |
| | | Kindergartens and primary schools (number) | 0.05 |
| | | Medical and health institutions (number) | 0.08 |
| | | Libraries, cultural stations and theaters (number) | 0.08 |

### 2.2.2. Data Standardization

In order to eliminate the influence of data dimension, this paper normalizes the dispersion of data. Where $X_i^*$ is the index data after standardization, and the value is between [0, 1]:

$$\text{Positive indicator}: \ X_i^* = \frac{(X_i - X_{min})}{(X_{max} - X_{min})} \tag{1}$$

$$\text{Negative indicator}: \ X_i^* = \frac{(X_{max} - X_i)}{(X_{max} - X_{min})}. \tag{2}$$

In order to reduce the impact of subjective factors on the final result of the evaluation based on personal experience, this paper uses the coefficient of variation weighting method to determine index weight, the specific formula is as follows,

$$V_i = \frac{\sigma_i}{x_i} \tag{3}$$

where $V_i$ is the coefficient of variation of the $i$ index, $\sigma_i$ is the standard deviation of the $i$ index, and $x_i$ is the average of the $i$ index. The weight of each indicator is:

$$W_i = \frac{V_i}{\sum_{i=1}^{n} V_i}. \tag{4}$$

### 2.2.3. Niche-Related Methods

(1) The degree of niche breadth

Niche breadth is also called niche width or niche size, which is the sum of the utilization degree of species to the environment and resources [48]. The greater the niche breadth, the stronger the survival adaptability. In this paper, the improved Shannon-Wiener index formula is used to calculate the niche width of towns. The calculation formula is,

$$B_i = -\sum_{j=1}^{s} P_{ij} log P_{ij} \qquad (5)$$

where, $B_i$ is the competitive niche width of town $i$, $P_{ij}$ is the product of utilization ratio and comprehensive weight of $j$ resources of town $i$ and $s$ is the number of indicators. The results of niche breadth were divided into five grades by the natural break point classification method. The larger the value, the larger the niche breadth.

(2) The degree of niche overlap

Niche overlap refers to the similarity between two different species in the connection of ecological factors, i.e., the utilization degree of two species to the same resource niche [7]. The greater the overlap value between the town niche, it indicates that there is a competition relationship between the towns for more resources. Limited resources will intensify the competition and conflict between towns. In this paper, Pianka model was used to measure the niche overlap of towns,

$$O_{ij} = \sum_{i}^{n} P_{ij} P_{ik} / \sqrt{\sum_{i}^{n} P_{ij}^2 \sum_{i}^{n} P_{ik}^2} \qquad (6)$$

where $P_{ij}$ and $P_{ik}$ represent the proportions of the $i$ resource used by the $j$ and $k$ species.

(3) The degree of niche coordination

The development of towns needs the coordination of environmental, economic and social subsystems. In order to measure the level of coordinated development of each subsystem of towns, this study is based on the coordination degree formula, which is improved as follows,

$$C_{wxy} = \left\{ \frac{f(w)g(x)h(y)}{\left[\frac{f(w)+g(x)+h(y)}{3}\right]^3} \right\}^3 \qquad (7)$$

where $f(w)$, $g(x)$, $h(y)$ are the environmental, economic niche and social niche breadth of the town respectively; The higher the coordination index, the higher the coordination degree of each sub-system of the town.

### 2.2.4. Spatial Analysis Methods

(1) Spatial autocorrelation analysis

The papers use global Moran's $I$ to measure the spatial correlation of ecological niches in each town and the degree of difference,

$$I = \frac{n}{\sum_i \sum_j w_{ij}} \frac{\sum_i \sum_j w_{ij}(x_i - x)(x_j - x)}{\sum_i (x_i - x)^2} \qquad (8)$$

where $n$ is the total number of towns, $W_{ij}$ is the weight matrix, $x_i$, $x_j$ is the variable of niche breadth in this paper for the spatial unit town $i$ and $j$; $I$ value range is $[-1, 1]$, 0> indicates that the space presents clustering characteristics, <0 indicates that the space presents dispersion characteristics and 0 represents random distribution [49].

The paper uses local spatial autocorrelation to further measure the spatial difference between a town and its surrounding areas and identifies the cold spot areas and hot spot areas:

$$Anselin\ Local\ Moran'I = Z_i \sum_{j=1}^{n} w_{ij} Z_j. \tag{9}$$

(2) Gravitational model

The research will explore the influence degree of the distance between regions on the niche of towns. Based on the niche breadth of towns, the gravity model formula is improved as follows,

$$G_{ij} = \frac{B_i \times B_j}{D_{ij}^2} \tag{10}$$

where $G_{ij}$ is the interaction gravity between $i$ and $j$; $B_i, B_j$ is the comprehensive niche breadth of $i$ and $j$, and $D$ is the Euclidean distance between the two towns.

(3) Pearson correlation coefficient analysis

Pearson correlation coefficient is used to reflect the degree of linear correlation between two niches. The $R$ value range is $[-1, 1]$. The greater the absolute value, the stronger the correlation [50]:

$$R = \sum_{i=1}^{n} \left( X_i - X \right) \left( Y_i - Y \right) / \sqrt{\sum_{i=1}^{n} \left( X_i - X \right)^2} \sqrt{\sum_{i=1}^{n} \left( Y_i - Y \right)^2}. \tag{11}$$

*2.3. Data Source*

The data needed for the study can be divided into three categories: yearbook data, POI data (point of interest) and remote sensing data. The data of the Yearbook are from China's County Statistical Yearbook (town volume), Zhejiang Province's third national agricultural census sub Town data, Jiangsu Province's third national agricultural census sub Town data, and statistical yearbooks of 19 relevant cities like Suzhou Statistical Yearbook (2017), respectively. POI data is geographic data with spatial coordinates, including schools, hospitals, banks, catering, entertainment, etc., which can reflect the distribution and supply of public facilities in the research area. The original data used in the study is mainly from Google maps in 2017. The remote sensing data in this study are water area, wetland area and normalized difference vegetation index, which are extracted from the website of the Department of Earth System Science, Tsinghua University, Beijing, China (http://data.ess.tsinghua.edu.cn/fromglc2017v1.html). PM2.5 is from National Meteorological Information Center (http://www.nmic.cn/). The data was obtained by calculating the average of the towns in 2017.

## 3. Results

*3.1. Towns Difference of Sub-Niche*

3.1.1. Niche Breadth Analysis

The environmental, economic and social niche breadth was calculated by Formula (5) for towns, see Figure 3. The environmental niche performed with the characteristics of "high value in the northeast, low value in the southwest". Due to the high PM2.5 and heavy industry pollution, the environmental niche in Taiizhou and Nantong city are relatively low. The higher environment quality of towns implies better environmental welfares for residents, especially human health. Generally, the economic and social niche changed oppositely with environmental niche. There are obvious differences in economic niche between the north and the south. The niche breadth of each town is not balanced and the towns around the Tai lake presented the best performance, e.g., Suzhou, Wuxi, Huzhou and Changzhou city. However, Quzhou, Taizhou and Lishui city had the worst performance. The economic strength of Jiangsu Province in the north is stronger than that in Zhejiang Province in the south. The robust performance of town industry makes the regional GDP

of Jiangsu Province rank the second in China. According to statistics, 204 of China's top 1000 towns are in Jiangsu and 161 are in Zhejiang in 2019 (https://top.askci.com/). Social niche is related to residents' social demands, e.g., food, housing, and transportation. The higher value of social niche means higher social welfares in terms of convenient, comfortable, and healthy social conditions. The social niche is roughly distributed the similarly as the economic niche. Results of Pearson correlation analysis conducted by SPSS 22.0 software are shown that the correlation between the social niche and the economic niche is the highest, with the coefficient of 0.560, followed by the correlation between the economic niche and the environmental niche (−0.144). The weak correlation between social niche and environmental niche is observed with the coefficient of only 0.002. It is noted that the economic niche positively affects the social niche to a certain extent and has a negative correlation with the environmental niche. This may imply that economic devel opment does not necessarily lead to environmental improve ment.

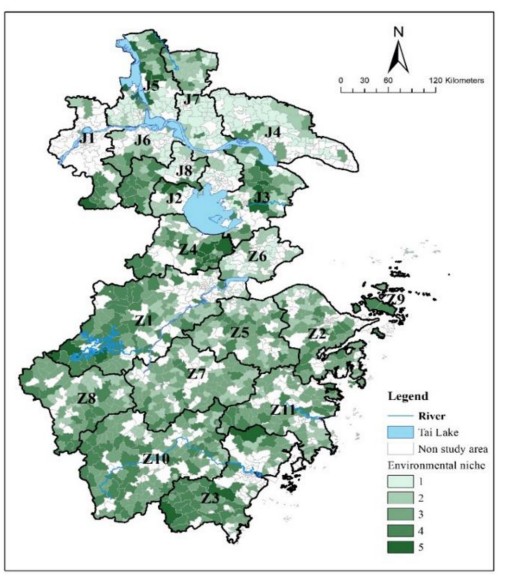

(**a**) Environmental niche

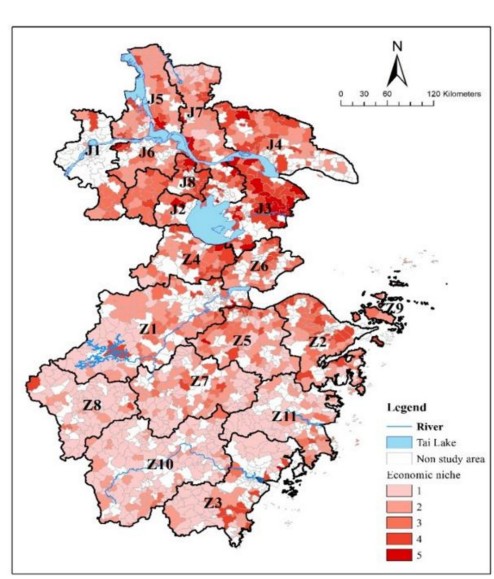

(**b**) Economic niche

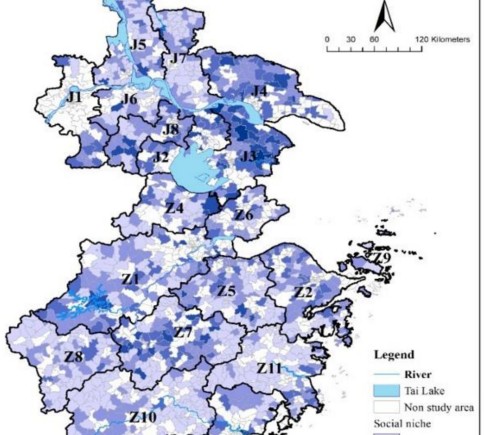

(**c**) Social niche

**Figure 3.** Town differences of environmental, economic, and social niche breadth.

### 3.1.2. Niche Coordination Development Analysis

The environmental, economic, and social coordination degree of towns was evaluated by Formula (7), see Figure 4. It can be seen that the coordination degree in Jiangsu Province is significantly higher than that in Zhejiang Province, with the characteristics of strong in the north and weak in the south. Spaces with larger value of niche breadth mean better coordination, e.g., Suzhou, Wuxi, Changzhou and Ningbo city. The differences in coordination degree are related with the relatively high environmental niche breadth and low economic and social niche breadth in Zhejiang province. Therefore, it is urgent to improve social welfares and economic performance in Zhejiang province. As shown by the barrel effect, the key to regional coordinated development lied in the improvement of the social and economic development.

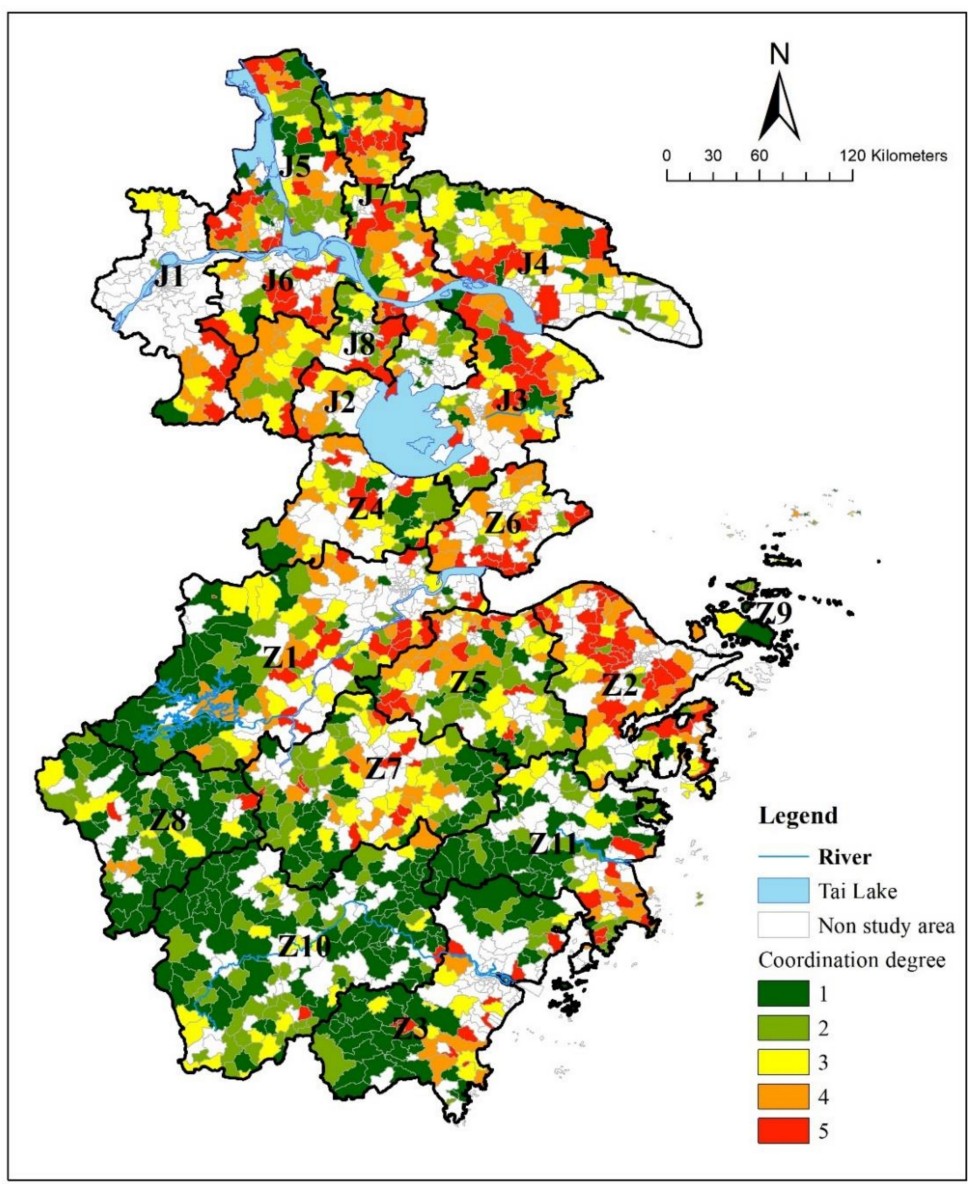

**Figure 4.** Town differences of environmental, economic, and social niche coordination degree.

### 3.2. Towns Differences of Comprehensive Niche

3.2.1. Spatial Correlation and Agglomeration of Comprehensive Niche

In the comprehensive niche, the interaction of environmental, economic, and social sub-systems, is crucial index to evaluate the overall development of towns. The spatial autocorrelation analysis, with the global Moran's I is 0.2511, reflecting the spatial distri-

bution characteristics of comprehensive niche, indicates a significant spatial correlation. The phenomenon of differentiation and agglomeration was observed in the town niche evolution. Spatial autocorrelation was further used to obtain spatial mapping for identify the differences of regional correlation. High-High cluster (HH) means that the town itself and surrounding towns have a large niche width, and Low-Low cluster (LL) is the opposite. High-Low outlier (HL) means that the town itself with a large niche width while the surrounding towns with the small niche width, and Low-High outlier (LH) is the opposite. Four types of agglomerations are displayed in Figure 5. HH agglomeration areas were mainly found in Suzhou, Wuxi and Huzhou city. LL agglomeration areas were mainly found in Taiizhou, Quzhou, Lishui and Taizhou city. These findings were consistent with spatial performance of niche breadth. Similar to the fact that species with the same attributes in biology are prone to form communities, in human society, towns with greater spatial autocorrelation and niche width tend to show strong spatial agglomeration. Towns with large niche breadth will drive the development of surrounding towns, while the agglomeration of towns with small niche breadth may worsen the developing situations and fall into the so-called Matthew effect.

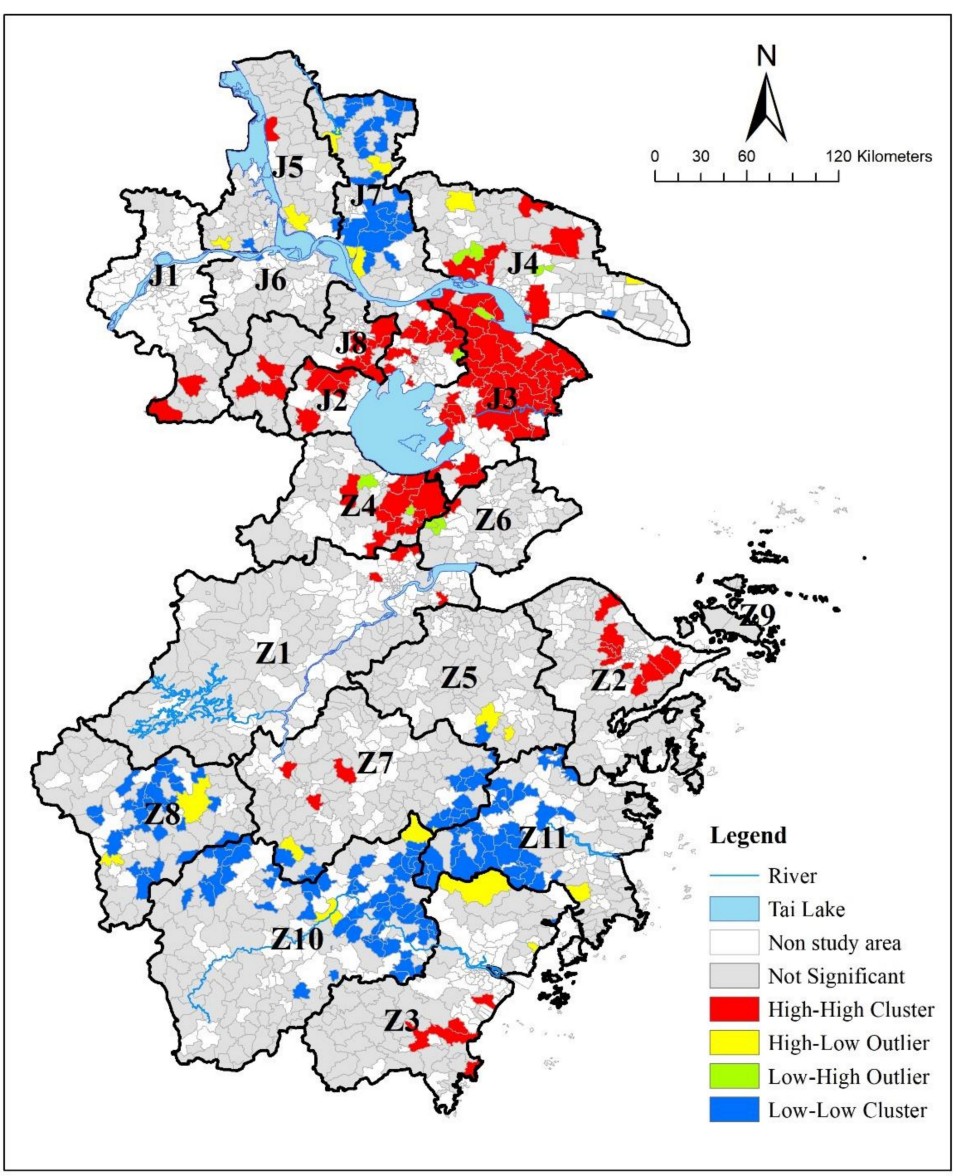

**Figure 5.** Spatial autocorrelation of comprehensive niche.

### 3.2.2. Co-Opetition Relationships of Comprehensive Niche

The niche overlap model was used to study the competitive relationships between two towns. There will be overlapping in niche quality when two towns compete for a certain resource. The overlapping niche of two towns is bound to competitive exclusion. According to Formula (6), 702,705 results of niche overlap are obtained for towns. The larger value the niche overlap, the smaller the resource dedicated to a town, and the stronger the competition. Figure 6 shows that the niche overlap of towns ranged from 0.01 to 0.99, of which the niche overlap over 0.9 accounted for 2.38% and those between 0.6–0.7 accounted for 20.08%.

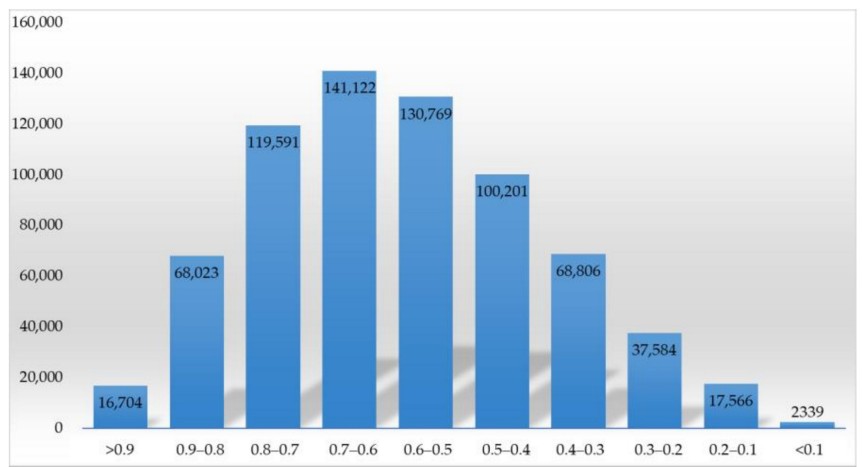

**Figure 6.** Niche overlap distribution of towns.

The overlap degree of prefecture city niche was calculated, see Figure 7. Lishui, Quzhou and Taizhou city have the highest degree of overlap with each other. It can also be found that the niche breadth of these cities is small. The reason is that these cities compete with limited ecological and economic resources in mountainous backgrounds, resulting in slow regional development. Cities with low niche breadth often display great niche overlap, which implies that greater niche overlap may hinder the development of cities.

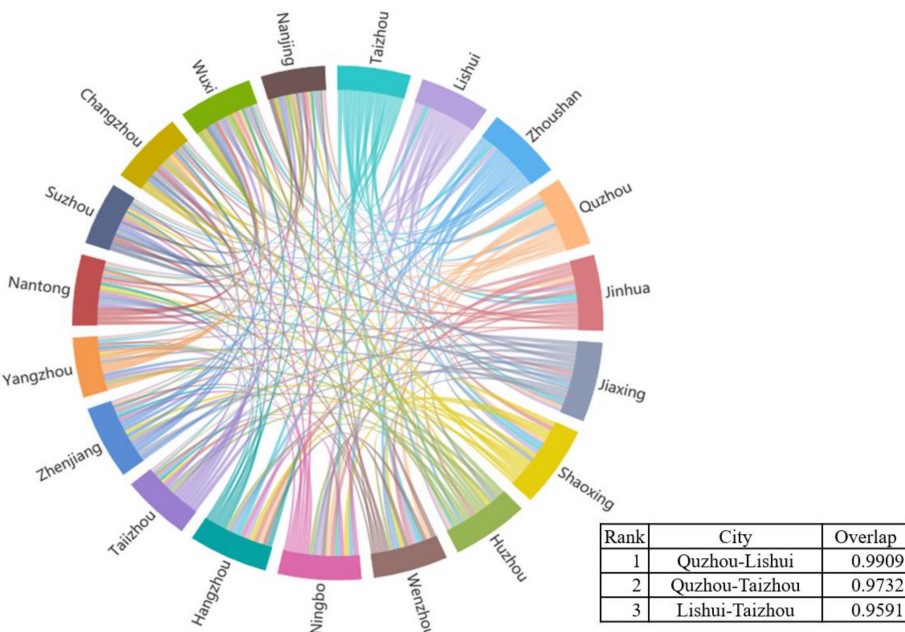

| Rank | City | Overlap |
|------|------|---------|
| 1 | Quzhou-Lishui | 0.9909 |
| 2 | Quzhou-Taizhou | 0.9732 |
| 3 | Lishui-Taizhou | 0.9591 |

**Figure 7.** Chord diagram of niche overlap of 19 prefecture cities (The richer the color of the lines, the lower the overlap).

Besides of the competitive relationship, the gravity model was employed to estimate the cooperation level and positive interaction among the selected cities. Figure 8 shows that Suzhou city had the strongest attraction to Wuxi city, followed by Changzhou and Zhenjiang city. Nanjing, Yangzhou, Taizhou and Zhenjiang city are also closely connected. In general, cities in Jiangsu Province demonstrated more close development than those in Zhejiang Province, which may due to economic differentiation, resources complementarity, and geographical accessibility. It shows that towns with smaller niche breadth are often subject to competition, while towns with larger niche breadth benefit from cooperative development.

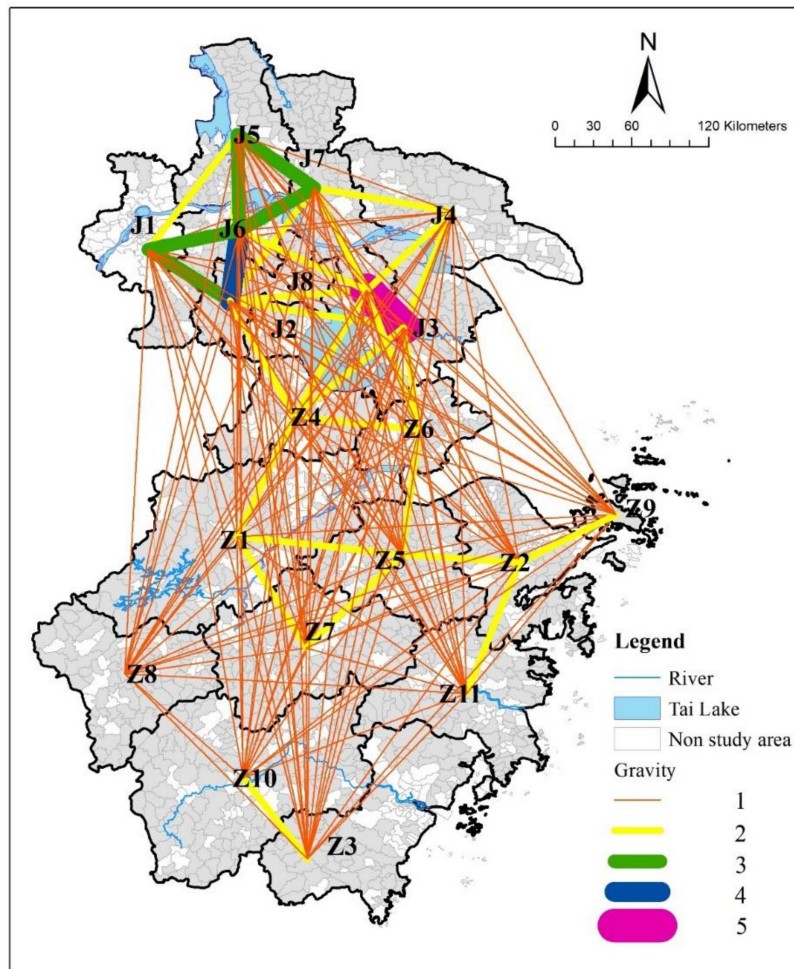

**Figure 8.** Urban differences of comprehensive niche gravity (1–5 is the gravity value. Bigger number means stronger gravity).

## 4. Discussion

### 4.1. Niche-Based Evaluation Framework

In this paper, a niche-based comprehensive evaluation framework is proposed for observing the spatial interaction and sustainability evolution of town system. Previous studies focused on animals, plants and other ecological fields to describe biodiversity and community phenomenon [34,35]. In this study, the concepts and tools of sociology, geography, and sustainability science can be incorporated into the conventional niche theory for depicting the niche co-opetition relationships and spatial transformation in 1186 towns in eastern China. Most of studies are based on the macroscale city, this provides new insights for the analysis of town scale [1,13,45]. The advent of niche-based framework will facilitate interdisciplinary exchanges among geography, sociology, landscape ecology and regional planning [51]. The niche-based framework blurs the divide between natural science and

social science and draws attention to the biological consequences of the built environment and socioeconomic structures underpinning urban life [1]. The results analyzed using the proposed framework could benefit to extraction of niche-related measures and pathways for enhancing competition and optimizing town patterns.

### 4.2. Regional Coordinated Development

The unbalanced development of environmental, economic, and social niche with obvious north-south characteristics are observed using the niche breadth index. The regional discrepancy of economic niche will drive people to gather in developed regions. Meanwhile, a series of adverse effects, such as hollowing settlements, labor shortage, and social instability, will encounter with the backward regions [21,51]. The social niche breadth indicates that large gaps remained in human well-being, such as medical, health, education, and infrastructure in various towns. The case study area we selected is the leading area for the development of Chinese towns, but it also faces the unbalanced development of society, economy and environment. China's early economic development was at the expense of environmental consumption, but it undermined the sustainable economic development and living welfares. Han et al. thought China's construction should pay more attention to the quality of people's lives rather than the economic development [52]. Jiang et al. also believed that it is necessary to strengthen the coordinated development of environment, economy, and society [13]. Exploring the complementary resource allocation and coordinated niche for the healthy development of Chinese towns has a long way to go.

### 4.3. Co-Opetition Relationships

This study reveals that the differentiation and agglomeration of towns with various niches will result in different developing state and trends. The development basis of the cities and towns follows the law of natural selection [53]. This highlighted that niche divergence play an important role in driving speciation in biology, and niche reorganization will foster adaptation and cooperation [16]. The towns with lager niche breadth and the smaller niche breadth all show agglomeration. This result was consistent with previous research [13]. The co-opetition analysis shows that the agglomeration in towns with small niche will not be conducive to the improvement of competitiveness. In contrast, towns with large niche takes full advantage of economic geography to cooperate together and grow strong. This performance is similar to the biological phenomenon, that is, once agglomerated, the richer the community, the stronger the ability to absorb resources [15]. However, this phenomenon often results in Matthew effect, which causes polarization and has a negative impact on the balanced regional pattern. The incoordination of the environmental-economic-society subsystem will also affect the development of towns. In the bucket effect, no matter how high a bucket is, the height of the inside water depends on the lowest wooden board. Therefore, the regional sustainable development cannot rely solely on the economic niche, leading to the pursuit of economic development and causing environmental imbalance [52]. For towns with small ecological niche, getting rid of the Barrel effect and the Matthew effect is a huge challenge in future development.

### 4.4. Suggestions and Strategies

The unbalanced, repetitive and low level of niche development, e.g., the barrel effect and the Matthew effect, is not conducive to the sustainable town development. To escape this dilemma, the following strategies should be considered in future development. (1) Niche separation strategy. In our study, the niche overlap degree of each town has a large span, which indicates different competition degrees among the selected towns [51]. In the conventional niche theory, in the case of resource saturation, there are usually enough differences between coexisting species. This difference will lead to the niche separation of coexisting species, thereby reducing the competitive intensity between species [5]. Thus, it is advisable to adopt functional niche separation in regional towns in the long term. The alternative developing measures for towns include strengthening development advantages,

building city brand [23], exploring characteristic development modes, and developing cutting-edge technologies. (2) Collaborative development strategy. The results show that the overlap degree between several cities is not high in Jiangsu Province. Due to the advantage of economic geography, these cities and towns have a close gravity, which benefits to coordinated development within and outside the region. Thus, in order to promote regional coordinated development, policy intervention is vital and effective [54,55]. Local governments need to cooperate closely to formulate overall development plans, including coordinating resource allocation, strengthening environmental collaboration, stopping local protectionism, and breaking institutional barriers, so as to realize the free flows of people, materials, capitals and information. (3) Symbiotic strategy. All creatures living with others survive through competition for limited resources and favorable environment, and symbiosis in sharing the ecological niche. This principle exists in both natural and human ecosystems, and is the main attribute in all communities [2,13]. Therefore, a successful city or town development should be the one maximizing its resource availability and optimizing its life strategy in order to adapt itself to make efficient use of its environment.

## 5. Conclusions

The ecological niche is a novel concept in sustainable science and a useful toolbox in analyzing competitive relationships for cities and towns. Some insights have been gathered in this exploratory study. The niche-based comprehensive evaluation framework is modified by conventional niche theory to identify niche dynamics and spatial differences of 1186 towns in the Yangtze River delta. The framework combines a three-layer niche-based evaluation index system, three niche-related methods and three spatial analysis methods. The proposed framework could help us understand the co-benefits and negative effects during the niche competition and cooperation. The results indicate that:

(1) From the perspective of niche breadth and spatial distribution, the distribution of niche shows obvious north-south difference and echelon distribution characteristics. The cities around the Tai Lake lead the entire study region. Moreover, there were obvious community differentiation characteristics.

(2) In terms of niche coordination, the coordination degree of Jiangsu Province is higher than that of Zhejiang Province. The higher the subsystem coordination degree, the better the town development.

(3) When it comes to the competition and cooperation in towns, Lishui and Quzhou have the highest niche overlap and the most obvious resource competition, while Suzhou and Wuxi have the largest urban attraction and more convenient cooperation. It shows that towns with poor ecological position are often subject to competition, while towns with good ecological position benefit from cooperative development.

(4) The harmonious, sustainable and healthy development of the region can be realized only when the ecological niche is coordinated and complementary among towns. Therefore, reasonable regulation of the town ecological niche plays an important role in the sustainable development of cities. It can take niche separation and collaborative development strategy to expand the functional niche of each town as much as possible.

In short, the concept of coordinated and sustainable environmental-social-economic development is now accepted. The ecological niches of complex ecosystems can better reflect the suitability of a region for various human activities, as well as the advantages and disadvantages of a resources, environment, society and economy, and provide more effective suggestions for the sustainable development of towns. Just as any case study, there are some limitations and possible improvements of future study. First, limited by the availability of data, the index selection in this study is insufficient. In the future, the indexes associating with big data and social text data could be improved the evaluation index system and measurement model of town ecological niche. Second, more time samples could be involved in observing the diversified long-term dynamics. Third, social network analysis (SNA), questionnaire, field interview could be incorporated into future study for improving methods system.

**Author Contributions:** D.L., conceptualization, data curation, writing-original draft, visualization. A.H., writing-review and editing. D.Y., formal analysis, writing-review and editing. J.L. (Jianyi Lin) and J.L. (Jiahui Liu), writing-original draft. All authors have read and agreed to the published version of the manuscript.

**Funding:** This research was supported by the National Natural Science Foundation of China (41371535), the Fundamental Research Funds for the Central Universities (SWU019047), the Ministry of Education of Humanities and Social Science Project (18YJA630039) and the Subsidized Project for Postgraduates' Innovative Fund in Scientific Research of Huaqiao University (18011121003).

**Conflicts of Interest:** The authors declare no conflict of interest.

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
