# Peer review of "Niche-Driven Socio-Environmental Linkages and Regional Sustainable Development"

_sustainability, doi:10.3390/su13031331_

Round 1

Reviewer 1 Report

A paper with interesting content, but needs certain improvements before proceeding further. I have provided detailed guidance in the reviewed pdf.

Among these issues, the following stand out:

  1. Reword your title to avoid existing duplication.
  2. Your abstract needs significant improvement and more clarity (see pdf)
  3. You intro should ensure a normal flow, with certain English language improvements required (see pdf)
  4. Ensure that you provide the source of all your maps/pictures etc.
  5. More importantly, in line 409 it is better to create a separate "subsection" and explain with more details the following strategies.
  6. (At your own decision) consider if "future work" should be included in your conclusions section

Good luck with your revision!

Author Response

Response to Reviewer 1 Comments

In new MS, we reorganize and rewrite most part of paper to make our paper logically readable. We hope the new MS has addressed the scientific basis with new reference criteria and all what you concern. Thank you for your constructive comments.

Point 1: Reword your title to avoid existing duplication. There is a "repetition" of the word ecological, consider to revise.

Response 1: After careful consideration, I changed the title to Niche-driven socio-environmental linkages and regional sustainable development.

Point 2: Your abstract needs significant improvement and more clarity (see pdf). (1) Sentence1: The changes in niche roles and functions caused by competition for survival resources are extending from natural science to social science. Not sure if "extending" is conveying the right message. Maybe replace with "having implications in various domains, with XXX and YYY standing out"? (2) Sentence2: A niche-based comprehensive evaluation framework, consisting of three-layer niche-based index system, niche-related methods and spatial analysis methods, which depicts the interactive relationships and spatial pattern of environmental, economic, and societal sub-system. Too long sentence, difficult for the non-experienced reader. Please reword/explain better. (3) Sentence3: The proposed framework can provide insights for understanding regional co-opetition relationship and regional transformation from the niche evolution and multi-elements allocation. "Vague statement". Please reword to convey the right message and relate to "future work" that is currently missing. (4) Too many uses of the term "niche" in keywords; please try to "trim" and ensure a better matching with the title.

Response 2: We followed your suggestions and recombined the abstract and keywords. Abstract: The changes in niche roles and functions caused by competition for survival resources are having implications in various domains, with natural science and social science standing out. Currently, expanding the ecological niche concept and its practical interpretation in the fields of social ecology, geography and sustainable science is becoming a crucial challenge. This paper based on niche theory to observe niche evolution and resulting socio-ecological effects of 1186 towns in 19 prefecture cities in Yangtze River delta. Results indicate that: Towns around the Taihu Lake displayed obvious spatial agglomeration, which was leading the development of the entire region. The town niche shows obvious characteristics of north-south differences and hierarchy distribution. The niche coordination degree of Jiangsu Province was higher than that of Zhejiang Province. The higher the subsystem coordination degree, the better the town development. Towns with poor ecological conditions are often subject to competition, while towns with better ecological conditions often benefit from cooperative development. The niche separation and collaboration could enhance niche competition of towns and cities in the region. The proposed framework can facilitate interdisciplinary exchanges among geography, sociology, landscape ecology and regional planning and provide insights for understanding regional co-opetition relationship and regional sustainable development.

Keywords: Niche; Spatial pattern; Co-opetition relationship; Regional regulation; Towns

Point 3: You intro should ensure a normal flow, with certain English language improvements required (see pdf).

Response 3: Thank you for your constructive comments. In the new introduction, we have revised all the problems you mentioned and reorganized the English language. You can see the line 30,31,35,37.

Point 4: Ensure that you provide the source of all your maps/pictures etc.

Response 4: We ensure that all maps and pictures are our original work. In the new manuscript, we improve the readability of Figures 1, 3, 4, 5, 8.

Point 5: More importantly, in line 409 it is better to create a separate "subsection" and explain with more details the following strategies.

Response 5: In the new manuscript, we reorganized the discussion subsection into 4 subsections, and concentrated all the strategies in one subsection. You can see the chapter 4.4 Suggestions and strategies.

The unbalanced, repetitive and low level of niche development, e.g., the barrel effect and the Matthew effect, is not conducive to the sustainable town development. To get out of this dilemma, the following strategies should be considered in future development. (1) Niche separation strategy. In our study, the niche overlap degree of each town has a large span, which indicates different competition degrees among the selected towns [51]. In the conventional niche theory, in the case of resource saturation, there are usually enough differences between coexisting species. This difference will lead to the niche separation of coexisting species, thereby reducing the competitive intensity between species [5]. Thus, it is advisable to adopt functional niche separation in regional towns in the long term. The alternative developing measures for towns include strengthening development advantages, building city brand [23], exploring characteristic development modes, and developing cutting-edge technologies. (2) Collaborative development strategy. Results show that the overlap degree between several cities is not high in Jiangsu Province. Due to the advantage of economic geography, these cities and towns have a close gravity, which benefits to coordinated development within and outside the region. Thus, in order to pro-mote regional coordinated development, policy intervention is vital and effective [54,55]. Local governments need to cooperate closely to formulate overall development plans, including coordinating resource allocation, strengthening environmental collaboration, stopping local protectionism, and breaking institutional barriers, so as to realize the free flows of people, materials, capitals and information. (3) Symbiotic strategy. All creatures living with others survive through competition for limited resources and favorable environment, and symbiosis in sharing the ecological niche. This principle exists in both natural and human ecosystems, and is the main attribute in all communities [2,13]. Thus, a successful city or town development should be the one maximizing its resource availability and optimizing its life strategy in order to adapt itself to make efficient use of its environment.

6.(At your own decision) consider if "future work" should be included in your conclusions section

Response 6: We followed your suggestions to add a future outlook in the conclusion section.

In short, the concept of coordinated and sustainable environmental-social-economic development is now accepted. The ecological niches of complex ecosystems can better reflect the suitability of a region for various human activities, as well as the advantages and disadvantages of a resources, environment, society and economy, and provide more effective suggestions for the sustainable development of towns. Just as any case study, there are some limitations and possible improvements of future study. First, limited by the availability of data, the index selection in this study is insufficient. In the future, the indexes associating with big data and social text data could be improved the evaluation index system and measurement model of town ecological niche. Second, more time samples could be involved in observing the diversified long-term dynamics. Third, social network analysis (SNA), questionnaire, field interview could be incorporated into future study for improving methods system.

Reviewer 2 Report

The reviewed article addresses a highly-interesting topic, namely the links between ecological niches and urban development. It is based on in-depth study, and the authors demonstrate the appropriate awareness of the literature. I tend to recommend this paper for acceptance, although see urgency of certain improvements (see below).

  • The title should reflect that this paper focuses on urban space and China. Is this paper really about regional transformation? In my opinion, it is about sustainable urban development!
  • Please, avoid numbering in the abstract.
  • You do not need to subdivide Introduction into sub-sections.
  • 3: all published sources of information should be cited properly and put to the list of references.
  • Which software is used to draw Fig. 7?
  • I cannot agree with Discussion with one reference. In addition to what you have done, I suggest to include to Discussion another sub-section comparing your findings with the results of some other studies (may be in the other countries) – the literature will be cited there.
  • Conclusions: please, state the perspectives for further research.
  • The writing is more or less clear, but, please, polish the language a bit else. For instance, the title of 3.1 is fully unclear.

Author Response

Response to Reviewer 2 Comments

Point 1: The title should reflect that this paper focuses on urban space and China. Is this paper really about regional transformation? In my opinion, it is about sustainable urban development!

Response 1: After careful consideration, I changed the title to Niche-driven socio-environmental linkages and regional sustainable development.

Point 2: Please, avoid numbering in the abstract.

Response 2: We followed your suggestions and deleted the numbers in the abstract.

Abstract: The changes in niche roles and functions caused by competition for survival resources are having implications in various domains, with natural science and social science standing out. Currently, expanding the ecological niche concept and its practical interpretation in the fields of social ecology, geography and sustainable science is becoming a crucial challenge. This paper based on niche theory to observe niche evolution and resulting socio-ecological effects of 1186 towns in 19 prefecture cities in Yangtze River delta. Results indicate that: Towns around the Taihu Lake displayed obvious spatial agglomeration, which was leading the development of the entire region. The town niche shows obvious characteristics of north-south differences and hierarchy distribution. The niche coordination degree of Jiangsu Province was higher than that of Zhejiang Province. The higher the subsystem coordination degree, the better the town development. Towns with poor ecological conditions are often subject to competition, while towns with better ecological conditions often benefit from cooperative development. The niche separation and collaboration could enhance niche competition of towns and cities in the region. The proposed framework can facilitate interdisciplinary exchanges among geography, sociology, landscape ecology and regional planning and provide insights for understanding regional co-opetition relationship and regional sustainable development.

Point 3: You do not need to subdivide Introduction into sub-sections.

Response 3: We followed your suggestions and reorganized the Introduction.

Point 4: all published sources of information should be cited properly and put to the list of references.

Response 4: We reviewed the literature, added some published literature and listed it in the list of references. You can find it in the introduction and discussion. Thank you for your suggestion.

Point 5: Which software is used to draw Fig. 7?

Response 5: Fig. 7 is a chord diagram showing the relationship between data. In this paper, it is made by a special drawing website (http://www.tubiaoxiu.com/).

Point 6: I cannot agree with Discussion with one reference. In addition to what you have done, I suggest to include to Discussion another sub-section comparing your findings with the results of some other studies (may be in the other countries) – the literature will be cited there.

Response 6: In the new manuscript, we have reorganized the discussion section and divided it into four parts (4.1. Niche-based evaluation framework; 4.2. Regional coordinated development; 4.3. Co-opetition relationships; 4.4. Suggestions and strategies). In particularly, it supplements the results of comparison with other studies. “In this paper, a niche-based comprehensive evaluation framework is proposed for observing the spatial interaction and sustainability evolution of town system. Previous studies focused on animals, plants and other ecological fields to describe biodiversity and community phenomenon [34,35]. In this study, the concepts and tools of sociology, geography, and sustainability science can be incorporated into the conventional niche theory for depicting the niche co-opetition relationships and spatial transformation in 1186 towns in eastern China. Most of studies are based on the macroscale city, this provides new insights for the analysis of town scale [1,13,45]. The case study area we selected is the leading area for the development of Chinese towns, but it also faces the unbalanced development of society, economy and environment. China's early economic development was at the expense of environmental consumption, but it undermined the sustainable economic development and living welfares. Han et al. thought China’s construction should pay more attention to the quality of people’s lives rather than the economic development [52]. Jiang et al. also believed that it is necessary to strengthen the coordinated development of environment, economy, and society [13].”

You can see the line 401-408,422-429,433-443 and so on.

Point 7: Conclusions: please, state the perspectives for further research.

Response 7: We followed your suggestions to add a future outlook in the conclusion section.

In short, the concept of coordinated and sustainable environmental-social-economic development is now accepted. The ecological niches of complex ecosystems can better reflect the suitability of a region for various human activities, as well as the advantages and disadvantages of a resources, environment, society and economy, and provide more effective suggestions for the sustainable development of towns. Just as any case study, there are some limitations and possible improvements of future study. First, limited by the availability of data, the index selection in this study is insufficient. In the future, the indexes associating with big data and social text data could be improved the evaluation index system and measurement model of town ecological niche. Second, more time samples could be involved in observing the diversified long-term dynamics. Third, social network analysis (SNA), questionnaire, field interview could be incorporated into future study for improving methods system.

Point 8: The writing is more or less clear, but, please, polish the language a bit else. For instance, the title of 3.1 is fully unclear.

Response 8: In new MS, we reorganize and rewrite part of paper to make our paper logically readable. We hope the new MS has addressed the scientific basis with new reference criteria and all what you concern. Thank you for your constructive comments. And the title of 3.1 was changed to Towns difference of sub-niche.

Reviewer 3 Report

The article is relevant and the topic is worthy of research. A higher level of academic scholarship is sometimes lacking in the introductory and scientific discussion sections. Therefore, I recommend a minor revision. If the authors are able to incorporate the suggested recommendations in a correct way, I believe that the article could be considered for publication in the journal. Here are the issues that authors need to address. I wish the authors luck with their manuscript.

REVIEW OF THE STATE OF THE ART

Authors carry out a literature survey, however, the bibliographic review is mainly aimed at conceptually explaining associated issues that other authors have studied. There is no true review of the state of the art from the methodological point of view that explains why there are currently various gaps in this field of research that justify the need to propose a new methodological framework. There are numerous alternative methodologies to the proposal raised by the authors (see https://www.mdpi.com/2071-1050/11/4/1039, https://www.sciencedirect.com/science/article/abs/pii/S0304380015002057 for example), the authors must cite alternative proposals like these and justify that their geostatistical approach improves the existing approaches.

SCIENTIFIC DISCUSSION 

The scientific discussion section is not very academic, as it is drafted in a schematic way, and more than a scientific discussion section it seems a schematic summary of the main points of the article. This section presents almost no bibliographic references. Authors should include bibliographic references to other previous studies in the area or on the methodology used, and analyze to what extent their results improve or corroborate / contradict said studies.

Author Response

Response to Reviewer 3 Comments

In new MS, we reorganize and rewrite most part of paper to make our paper logically readable. We hope the new MS has addressed the scientific basis with new reference criteria and all what you concern. Thank you for your constructive comments.

Point 1: REVIEW OF THE STATE OF THE ART

Authors carry out a literature survey, however, the bibliographic review is mainly aimed at conceptually explaining associated issues that other authors have studied. There is no true review of the state of the art from the methodological point of view that explains why there are currently various gaps in this field of research that justify the need to propose a new methodological framework. There are numerous alternative methodologies to the proposal raised by the authors (see https://www.mdpi.com/2071-1050/11/4/1039, https://www.sciencedirect.com/science/article/abs/pii/S0304380015002057 for example), the authors must cite alternative proposals like these and justify that their geostatistical approach improves the existing approaches.

Response 1: We followed your suggestions to reorganize the research on methods. See the line 39-44, 96-106 in new MS.

“During recent years, it has been noticed that rapid urbanization is accompanied by complicated challenges in terms of materials, energy and information issues. These challenges need to be comprehensively investigated by observing multiple factors [5-7]. The early research methods of socioecological system mainly include evaluation methods such as analytic hierarchy process, ecological footprint analysis and grey model [8-10]. In the later stage, scholars focused on the application of models and theories, such as DPSIR model [11], Bayesian network [12], full permutation polygon synthetic indicator [13], GIS participation mapping [14], niche and so on. Rooted in the fields of ecology and biogeography, ecological niche refers to the spaces and roles of species in biological community [15] which indicates the interactive relation-ships between a certain species with the outside environment [7,16-18]. From the perspective of research methods, the concept of niche is abstract and fuzzy. What we can learn from it is some quantitative indicators such as niche width, niche overlap, niche volume and niche dimension. Niche breadth and niche overlap are the most important quantitative indicators. Levins was the first to put forward the measurement formula of niche width, and also have Schoener, Hurlbert, Smith, etc [43]. The formulas of niche overlap are curve average method, symmetry α method [44] and so on. At present, the theory of niche is still in continuous development, and various measurement formulas are constantly improving, especially in the quantitative calculation of niche width. Each formula has its advantages and disadvantages. It can be predicted that with the continuous improvement of niche theory, it will inject more fresh blood into the theory and method of urban research.”

Point 2: SCIENTIFIC DISCUSSION

The scientific discussion section is not very academic, as it is drafted in a schematic way, and more than a scientific discussion section it seems a schematic summary of the main points of the article. This section presents almost no bibliographic references. Authors should include bibliographic references to other previous studies in the area or on the methodology used, and analyze to what extent their results improve or corroborate / contradict said studies.

Response 2: In the new manuscript, we have reorganized the discussion section and divided it into four parts (4.1. Niche-based evaluation framework; 4.2. Regional coordinated development; 4.3. Co-opetition relationships; 4.4. Suggestions and strategies). In particularly, it supplements the results of comparison with other studies. “In this paper, a niche-based comprehensive evaluation framework is proposed for observing the spatial interaction and sustainability evolution of town system. Previous studies focused on animals, plants and other ecological fields to describe biodiversity and community phenomenon [34,35]. In this study, the concepts and tools of sociology, geography, and sustainability science can be incorporated into the conventional niche theory for depicting the niche co-opetition relationships and spatial transformation in 1186 towns in eastern China. Most of studies are based on the macroscale city, this provides new insights for the analysis of town scale [1,13,45]. The case study area we selected is the leading area for the development of Chinese towns, but it also faces the unbalanced development of society, economy and environment. China's early economic development was at the expense of environmental consumption, but it undermined the sustainable economic development and living welfares. Han et al. thought China’s construction should pay more attention to the quality of people’s lives rather than the economic development [52]. Jiang et al. also believed that it is necessary to strengthen the coordinated development of environment, economy, and society [13].”

You can see the line 401-408,422-429,433-443 and so on.

Reviewer 4 Report

The manuscript “Ecological niche-driven socio-ecological linkages and regional transformation" is an interesting contribution to the field of sustainable science. In my opinion, this paper suits well into the scope of the Journal. It is methodologically well written, except for minor inaccuracies.

My comments relate mainly to the edition of the work and the way of presenting the results. I hope the consideration of some of them will make the manuscript better readable and more explicit regarding the merits it already has.

However, I have some suggestions that may improve the article.

SPECIFIC COMMENTS:

I have noticed many editing errors. One of them, very common at this work, is lack of spaces before references to literature. This editing part requires a significant improvement.

Abstract:

  • Abstract – The abstract doesn’t contain specific information about research aim and research methods used in article. It mainly includes the results of work

Keywords:

  • L28 – Please find such words which are not in the title, this way search engines of the web will find your manuscript with higher probability.

Introduction:

  • The introduction does not strongly indicate innovation in the conducted research. I encourage the authors to highlight there the innovative solutions developed in this study

Materials and Methods:

  • L135: Please explain what the GDP abbreviation stands for
  • In chapter 2.2.1. there is no justification as to why these and no other indicators were selected for each Sub-Target Layers. Please justify or explain such a selection
  • L201: Please explain why the results of niche breadth were divided into five grades and why you used the natural break point classification method
  • Various formatting used to explain the components of the formula. Sometimes italics is used (like in L240) and sometimes not (like in 200), and sometimes it is mixed up (like in L228). Please unify it.
  • L249: The POI abbreviation should be explained here (in L249), where it appears for the first time in the text, not in line 253

Results

  • L287: The title of the figure should be placed below the figure
  • Figure 3 is of poor quality. I suggest making it more readable
  • Figure 4 and 5: The names of the cities in the figures are difficult to read. What is more Figure 5 is of poor quality - the boundaries of the regions (especially those on a red background) are blurred
  • Figure 8: low quality of this Figure. Gravity lines marked with number 1 is unreadable. Please improve the quality of this figure

Discussion:

  • In Discussion section, the authors should discuss the results of the research and compare them with other studies. In the discussion should be cited appropriate literature. Try to discuss the results with relation to other Asian or European countries and cities. I suggest rewriting the discussion and focusing on comparing the results with other cases.

Author Response

Response to Reviewer 4 Comments

In new MS, we reorganize and rewrite most part of paper to make our paper logically readable. We hope the new MS has addressed the scientific basis with new reference criteria and all what you concern. Thank you for your constructive comments.

Point 1: I have noticed many editing errors. One of them, very common at this work, is lack of spaces before references to literature. This editing part requires a significant improvement.

Response 1: Thank you for your suggestion. It has been revised in the new MS.

Point 2: Abstract – The abstract doesn’t contain specific information about research aim and research methods used in article. It mainly includes the results of work.

Response 2: In the new MS, we have deleted the introduction of some research methods.

Abstract: The changes in niche roles and functions caused by competition for survival resources are having implications in various domains, with natural science and social science standing out. Currently, expanding the ecological niche concept and its practical interpretation in the fields of social ecology, geography and sustainable science is becoming a crucial challenge. This paper based on niche theory to observe niche evolution and resulting socio-ecological effects of 1186 towns in 19 prefecture cities in Yangtze River delta. Results indicate that: Towns around the Taihu Lake displayed obvious spatial agglomeration, which was leading the development of the entire region. The town niche shows obvious characteristics of north-south differences and hierarchy distribution. The niche coordination degree of Jiangsu Province was higher than that of Zhejiang Province. The higher the subsystem coordination degree, the better the town development. Towns with poor ecological conditions are often subject to competition, while towns with better ecological conditions often benefit from cooperative development. The niche separation and collaboration could enhance niche competition of towns and cities in the region. The proposed framework can facilitate interdisciplinary exchanges among geography, sociology, landscape ecology and regional planning and provide insights for understanding regional co-opetition relationship and regional sustainable development.

Point 3: Keywords: L28 – Please find such words which are not in the title, this way search engines of the web will find your manuscript with higher probability.

Response 3: We change the keyword to Niche; Spatial pattern; Co-opetition relationship; Regional regulation; Towns.

Point 4: Introduction: The introduction does not strongly indicate innovation in the conducted research. I encourage the authors to highlight there the innovative solutions developed in this study.

Response 4: We have organized a new introduction, and the innovation points are put at the end. You can see line 122-154.

Point 5: L135: Please explain what the GDP abbreviation stands for.

Response 5: Gross domestic product (GDP) is an important indicator to measure the overall economic situation of a region, which has been supplemented in the study.

Point 6: In chapter 2.2.1. there is no justification as to why these and no other indicators were selected for each Sub-Target Layers. Please justify or explain such a selection

Response 6: Town is essentially a complex ecosystem, the environment, resource and social system as supporting factor. Thus, analysis of niche involves many elements such as environment, economy and society [2,13]. Towns are very small in China's administrative di-visions, resulting in incomplete data. The index system of this study is based on the availability of data, combined with the foreign and domestic niche and sustainable development index system. According to different town functional modules and incorporating the system of suitable town indices, the target layer is the comprehensive ecosystem, the sub-target layer includes the niches of the environment, economy, and society, and the indicator layer comprises the individual indicators for the corresponding niches. The environmental niche reflects the advantages and disadvantages of the region in resource endowments in and natural ecosystem protection and restoration, includes water area, wetland area, normalized difference vegetation index, and Particulate Matter 2.5 (PM2.5). The economic niche indicates the advantages and disadvantages of the region in terms of economic developments, includes 4 indicators that reflect the economic scale and eco-nomic structure of the region. The social niche reflects the advantages and disadvantages of public facilities and welfare in a region, includes 6 indicators that reflect the advantages and disadvantages of a region regarding education, medical service, health, social security, quality of life, and rural living facilities.

Point 7: L201: Please explain why the results of niche breadth were divided into five grades and why you used the natural break point classification method.

Response 7: Each town has a niche value, and there are 1186 in this study. In order to compare the results of many towns, according to previous studies, the natural break point classification method is used to divide them into 5 categories. Compared with the division into 3, 7, 9, etc., 5 is more moderate. This study is to compare the development of towns in the region, so as long as the scale is divided, all towns can be compared horizontally.

Point 8: Various formatting used to explain the components of the formula. Sometimes italics is used (like in L240) and sometimes not (like in 200), and sometimes it is mixed up (like in L228). Please unify it. L249: The POI abbreviation should be explained here (in L249), where it appears for the first time in the text, not in line 253

Response 8: The format of the formula has been unified in the new MS, and the abbreviation of POI has been revised. Thank you for your suggestions.

Point 9: Results

L287: The title of the figure should be placed below the figure

Figure 3 is of poor quality. I suggest making it more readable

Figure 4 and 5: The names of the cities in the figures are difficult to read. What is more Figure 5 is of poor quality - the boundaries of the regions (especially those on a red background) are blurred?

Figure 8: low quality of this Figure. Gravity lines marked with number 1 is unreadable. Please improve the quality of this figure.

Response 9: We replaced city names with an invented IDs, for example J1 - Nanjing city, Jiansu province. Enter these IDs in Figure 1, 3, 4, 5, 8 and Table 1. The city boundaries in Figure 3.4.5.8 have been bolded and blackened and improved the readability of Figure 8.

Point 10: Discussion:

In Discussion section, the authors should discuss the results of the research and compare them with other studies. In the discussion should be cited appropriate literature. Try to discuss the results with relation to other Asian or European countries and cities. I suggest rewriting the discussion and focusing on comparing the results with other cases.

Response 10: In the new manuscript, we have reorganized the discussion section and divided it into four parts (4.1. Niche-based evaluation framework; 4.2. Regional coordinated development; 4.3. Co-opetition relationships; 4.4. Suggestions and strategies). In particularly, it supplements the results of comparison with other studies. “In this paper, a niche-based comprehensive evaluation framework is proposed for observing the spatial interaction and sustainability evolution of town system. Previous studies focused on animals, plants and other ecological fields to describe biodiversity and community phenomenon [34,35]. In this study, the concepts and tools of sociology, geography, and sustainability science can be incorporated into the conventional niche theory for depicting the niche co-opetition relationships and spatial transformation in 1186 towns in eastern China. Most of studies are based on the macroscale city, this provides new insights for the analysis of town scale [1,13,45]. The case study area we selected is the leading area for the development of Chinese towns, but it also faces the unbalanced development of society, economy and environment. China's early economic development was at the expense of environmental consumption, but it undermined the sustainable economic development and living welfares. Han et al. thought China’s construction should pay more attention to the quality of people’s lives rather than the economic development [52]. Jiang et al. also believed that it is necessary to strengthen the coordinated development of environment, economy, and society [13].”

You can see the line 401-408,422-429,433-443 and so on.

Reviewer 5 Report

The manuscript concerns about study of microscale analysis of functions and roles of cities and towns based on ecological niche theory.

Most of the visualizations in the form presented for review cannot be accepted. In the text of the paper, you use SI units (meters), but the Figures are described using miles. SI Units should be used.

City names interfere with the visualization of results in Figures 3, 4, 5 and 8. I suggest replacing city names with an invented IDs, for example J1 - Nanjing city, Jiansu province. Enter these IDs in Figure 1, 3, 4, 5, 8 and Table 1.

Add the citys boundaries in Figure 3, 4, 5 and 8.

Is it necessary to use towns boundaries to represent gravity value in Figure 8? Correct the thickness of the lines to correct Figure 8 or just use different colors to denote the gravity value.

Correct the quality of presentation results in Figure 3.

The manuscript needs to be improved. In the text you mention 8 times that you analyse 1186 towns. Is it necessary? There are many editorial errors in the text, e.g. ‘the the’ (line 280), abbreviations should be explained the first time they appear in the text (e.g. see line 249 and 253), etc.

Determine for each city what percentage of the city's area does not take part in the analysis (no data areas) and give these values also in km2. Does the lack of data from these areas (the spatial discontinuity) affect the results of the analysis? If so, in what way or at what level?

In line 124 you mention 2016, while on line 135 you only provide information for 2019. Complete the paper also with information about 2016 for Zhejiang and Jiangsu provinces. More, you use data from 2017, why do you mention 2016 in line 124?

In the Methods you mention that you use 14 factors (line 153). The proposed method should be universal. Should 14 indicators always be used in this method? Specify the minimum number of indicators which should be used.

Lines 166-168 partially repeat information from lines 151-153. I sugest to remove lines 166-168. Move explaining of dividing into three levels in the case of niche-based evaluation index system to Methods.

The research used data from different moments of time, the Yearbook and Google maps data for 2017 and the remote sensing data from 2015. In my opinion, the use of data from different moments in time may reduce the quality of information provided by proposed analysis. The authors' comment in this regard is recommended.

Expand the paper about the quality of the data used, especially points of interest. Can the number of objects determined on the basis of Google maps be incorrect?  Mapping resolution of the remote sensing data used and the percentage of areas with code 120-Cloud (if any) should be specified.

Add to Discussion more existing studies in this area or similar cities/towns.

Author Response

Response to Reviewer 5 Comments

In new MS, we reorganize and rewrite most part of paper to make our paper logically readable. We hope the new MS has addressed the scientific basis with new reference criteria and all what you concern. Thank you for your constructive comments.

Point 1: Most of the visualizations in the form presented for review cannot be accepted. In the text of the paper, you use SI units (meters), but the Figures are described using miles. SI Units should be used. City names interfere with the visualization of results in Figures 3, 4, 5 and 8. I suggest replacing city names with an invented IDs, for example J1 - Nanjing city, Jiansu province. Enter these IDs in Figure 1, 3, 4, 5, 8 and Table 1. Add the city boundaries in Figure 3, 4, 5 and 8. Is it necessary to use towns boundaries to represent gravity value in Figure 8? Correct the thickness of the lines to correct Figure 8 or just use different colors to denote the gravity value. Correct the quality of presentation results in Figure 3.

Response 1: We followed your suggestions and used SI units (meters or kilometers),city names replace with Jn(n=1,2,∙∙∙n) and Zn(n=1,2,∙∙∙n), as shown in Figure 1, 3, 4, 5, 8 and Table 1. The city boundaries in Fig. 3.4.5.8 have been bolded and blackened. Figure 8 if only used different colors to display, too many line segments with the gravity value of 1 will cover up the gravity value of other number, which is not conducive to the display of the results in Figure 8. Therefore, different thicknesses and colors are used to represent them together. Figure 3 has been redrawn.

Point 2: The manuscript needs to be improved. In the text you mention 8 times that you analyse 1186 towns. Is it necessary? There are many editorial errors in the text, e.g. ‘the the’ (line 280), abbreviations should be explained the first time they appear in the text (e.g. see line 249 and 253), etc.

Response 2: In the new MS, we have deleted some of the description of 1186 towns, and the errors in lines 249, 253 and 280 have been corrected. You can see the line 284, 288 and 330.

Point 3: Determine for each city what percentage of the city's area does not take part in the analysis (no data areas) and give these values also in km2. Does the lack of data from these areas (the spatial discontinuity) affect the results of the analysis? If so, in what way or at what level?

Response 3: The area not analyzed is the urban area and is not within the scope of this study. This paper is a horizontal comparison of town level. The area without data does not belong to town, the size of their area has nothing to do with this article, so it does not affect the analysis results.

Point 4: In line 124 you mention 2016, while on line 135 you only provide information for 2019. Complete the paper also with information about 2016 for Zhejiang and Jiangsu provinces. More, you use data from 2017, why do you mention 2016 in line 124?

Response 4: 2019 data is replaced with 2017 data in the new MS, you can see the chapter 2.1. and table 1. The 2019 data was used previously because this section is an introduction to the study area, so used the latest data. In order to make the full text one before and after, all the errors in the data year in this article have been revised. Thank you for your suggestions.

Point 5: In the Methods you mention that you use 14 factors (line 153). The proposed method should be universal. Should 14 indicators always be used in this method? Specify the minimum number of indicators which should be used.

Response 5: This method does not always use 14 indicators. Towns are very small in China's administrative divisions, resulting in incomplete data. The index system of this study is based on the availability of data, combined with the foreign and domestic niche and sustainable development index system. This study tried its best to obtain representative data, which is mentioned in the limitations of the conclusion of this article. In the future, the more complete the indicators, the higher the credibility of the research results.

Point 6: Lines 166-168 partially repeat information from lines 151-153. I suggest to remove lines 166-168. Move explaining of dividing into three levels in the case of niche-based evaluation index system to Methods.

Response 6: We followed your suggestion and deleted lines 166-168. The explaining of dividing into three levels in the case of niche-based evaluation index system was added to chapter 2.2.1.

Town is essentially a complex ecosystem, the environment, resource and social system as supporting factor. Thus, analysis of niche involves many elements such as environment, economy and society [2,13]. Towns are very small in China's administrative di-visions, resulting in incomplete data. The index system of this study is based on the availability of data, combined with the foreign and domestic niche and sustainable development index system. According to different town functional modules and incorporating the system of suitable town indices, the target layer is the comprehensive ecosystem, the sub-target layer includes the niches of the environment, economy, and society, and the indicator layer comprises the individual indicators for the corresponding niches. The environmental niche reflects the advantages and disadvantages of the region in resource endowments in and natural ecosystem protection and restoration, includes water area, wetland area, normalized difference vegetation index, and Particulate Matter 2.5 (PM2.5). The economic niche indicates the advantages and disadvantages of the region in terms of economic developments, includes 4 indicators that reflect the economic scale and eco-nomic structure of the region. The social niche reflects the advantages and disadvantages of public facilities and welfare in a region, includes 6 indicators that reflect the advantages and disadvantages of a region regarding education, medical service, health, social security, quality of life, and rural living facilities.

Point 7: The research used data from different moments of time, the Yearbook and Google maps data for 2017 and the remote sensing data from 2015. In my opinion, the use of data from different moments in time may reduce the quality of information provided by proposed analysis. The authors' comment in this regard is recommended. Mapping resolution of the remote sensing data used and the percentage of areas with code 120-Cloud (if any) should be specified.

Response 7: The remote sensing data in this study are water area, wetland area and normalized difference vegetation index, which are extracted from the website of the Department of Earth System Science, Tsinghua University, China (http://data.ess.tsinghua.edu.cn/fromglc2017v1.html). PM2.5 is from National Meteorological Information Center(http://www.nmic.cn/). The data was obtained by calculating the average of the towns in 2017.

Point 8: Expand the paper about the quality of the data used, especially points of interest. Can the number of objects determined on the basis of Google maps be incorrect?

Response 8: The original data is mainly crawled from Google Maps, which is the 2017 POI data of Jiangsu Province and Zhejiang Province. After deduplication and screening, we finally selected 414110 data. Because the crawling steps and entries of POI data in all towns are the same, there is a certain reference value in horizontal comparison.

Point 9: Add to Discussion more existing studies in this area or similar cities/towns.

Response 9: In the new manuscript, we have reorganized the discussion section and divided it into four parts (4.1. Niche-based evaluation framework; 4.2. Regional coordinated development; 4.3. Co-opetition relationships; 4.4. Suggestions and strategies). In particularly, it supplements the results of comparison with other studies. “In this paper, a niche-based comprehensive evaluation framework is proposed for observing the spatial interaction and sustainability evolution of town system. Previous studies focused on animals, plants and other ecological fields to describe biodiversity and community phenomenon [34,35]. In this study, the concepts and tools of sociology, geography, and sustainability science can be incorporated into the conventional niche theory for depicting the niche co-opetition relationships and spatial transformation in 1186 towns in eastern China. Most of studies are based on the macroscale city, this provides new insights for the analysis of town scale [1,13,45]. The case study area we selected is the leading area for the development of Chinese towns, but it also faces the unbalanced development of society, economy and environment. China's early economic development was at the expense of environmental consumption, but it undermined the sustainable economic development and living welfares. Han et al. thought China’s construction should pay more attention to the quality of people’s lives rather than the economic development [52]. Jiang et al. also believed that it is necessary to strengthen the coordinated development of environment, economy, and society [13].”

You can see the line 401-408,422-429,433-443 and so on.

Round 2

Reviewer 4 Report

Thanks to the authors of the manuscript for the answers to my questions and correction applied after the suggestions of all rewievers. I am satisfied with these changes and responses.

Author Response

We greatly appreciate your help in improving this article.

Reviewer 5 Report

Thanks to the authors of the manuscript for the answers to my questions and correction applied after the suggestions of all rewievers. I am satisfied with these changes, however, have noticed some issues that still need to be improved and explained.

According to your answer: 'The area without data does not belong to town, the size of their area has nothing to do with this article, so it does not affect the analysis results.' In connection with the above, I suggest not to use the term 'No data' in the figures. This term suggests that this area is disregarded due to a lack of data and not a methodology that precludes the analysis of urban area (In your answer, you notice: 'The area not analyzed is the urban area and is not within the scope of this study.'). 

In my review, I noted that the manuscript used 2015 remote-sensing data and asked for comment. An earlier version of the manuscript contained the sentence 'The remote sensing data was obtained from the web-site of the Department of Earth System Science, Tsinghua University, China (http://data.ess.tsinghua.edu.cn/fromglc2015_ v1.html)'. You did not make this point clear in your answer.

In this revision of the manuscript, you replaced this sentence with the following sentence, where you deleted the year of data: 'The remote sensing data was obtained from the website of the Department of Earth System Science, Tsinghua University, China (http://data.ess.tsinghua.edu.cn/)'.

In addition, in your answer to me not disclosed in the text of the revised manuscript, you provide: 'The remote sensing data in this study are water area, wetland area and normalized difference vegetation index, which are extracted from the website of the Department of Earth System Science, Tsinghua University, China (http : //data.ess.tsinghua.edu.cn/fromglc2017v1.html)' and you declare that you used remote-sensing date from 2017.

Reliably declare the years of the remote-sensing data used in the study in the manuscript text.

Author Response

Response to Reviewer 5 Comments

Point 1: According to your answer: 'The area without data does not belong to town, the size of their area has nothing to do with this article, so it does not affect the analysis results.' In connection with the above, I suggest not to use the term 'No data' in the figures. This term suggests that this area is disregarded due to a lack of data and not a methodology that precludes the analysis of urban area (In your answer, you notice: 'The area not analyzed is the urban area and is not within the scope of this study.').

Response 1: Thank you for your suggestion. It was convenient when using the expression ‘no data’, but it did cause misunderstanding which was not the original intention of the article. The use of such expression will be carefully chosen to achieve more precise and realistic description. Thank you again for your suggestion. All charts and expressions in this paper have been revised. You can see the Figure1,3,4,5,8 and line 151-152.

Point 2: In my review, I noted that the manuscript used 2015 remote-sensing data and asked for comment. An earlier version of the manuscript contained the sentence 'The remote sensing data was obtained from the web-site of the Department of Earth System Science, Tsinghua University, China (http://data.ess.tsinghua.edu.cn/fromglc2015_ v1.html)'. You did not make this point clear in your answer.

In this revision of the manuscript, you replaced this sentence with the following sentence, where you deleted the year of data: 'The remote sensing data was obtained from the website of the Department of Earth System Science, Tsinghua University, China (http://data.ess.tsinghua.edu.cn/)'.

In addition, in your answer to me not disclosed in the text of the revised manuscript, you provide: 'The remote sensing data in this study are water area, wetland area and normalized difference vegetation index, which are extracted from the website of the Department of Earth System Science, Tsinghua University, China (http : //data.ess.tsinghua.edu.cn/fromglc2017v1.html)' and you declare that you used remote-sensing date from 2017.

Reliably declare the years of the remote-sensing data used in the study in the manuscript text.

Response 2: Firstly, the remote-sensing data is indeed from 2017. However, the link was copied wrongly when writing the source due to negligence and this mistake was not noticed. Secondly, as you mentioned, contents that are not in the main text are disclosed in my reply. In fact, many details are clearly described in the early stages of the writing process. But many details have been deleted in the process of repeated modification. At that time, we thought that it was enough to write until (http://data.ess.tshunghua.edu.cn/), and it did not need to be very detailed. Therefore, we did not explain all the information about the data source in the article. We would like to thank you once again for your guidance. We have learned a lot from this revision and it is also beneficial to the future scientific research. The year of the remote-sensing data used has been further stated in the article. You can see the line 276-281.